# Review of the Interactions between Conventional Cementitious Materials and Heavy Metal Ions in Stabilization/Solidification Processing

**DOI:** 10.3390/ma16093444

**Published:** 2023-04-28

**Authors:** Jingjing Liu, Dongbiao Wu, Xiaohui Tan, Peng Yu, Long Xu

**Affiliations:** 1School of Resource and Environmental Engineering, Hefei University of Technology, Hefei 230009, China; tanxh@hfut.edu.cn (X.T.); xulong_2005@hfut.edu.cn (L.X.); 2Anhui Urban Construction Design Institute Corp., Ltd., Hefei 230051, China; wudb0425@163.com (D.W.); pengyu0213@126.com (P.Y.)

**Keywords:** heavy metal contamination, stabilization/solidification, remediation mechanisms, ion immobilization, interference impacts

## Abstract

In the past few decades, solidification/stabilization (S/S) technology has been put forward for the purpose of improving soil strength and inhibiting contaminant migration in the remediation of heavy metal-contaminated sites. Cement, lime, and fly ash are among the most common and effective binders to treat contaminated soils. During S/S processing, the main interactions that are responsible for improving the soil’s behaviors can be summarized as gelification, self-hardening, and aggregation. Currently, precipitation, incorporation, and substitution have been commonly accepted as the predominant immobilization mechanisms for heavy metal ions and have been directly verified by some micro-testing techniques. While replacement of Ca^2+^/Si^4+^ in the cementitious products and physical encapsulation remain controversial, which is proposed dependent on the indirect results. Lead and zinc can retard both the initial and final setting times of cement hydration, while chromium can accelerate the initial cement hydration. Though cadmium can shorten the initial setting time, further cement hydration will be inhibited. While for mercury, the interference impact is closely associated with its adapted anion. It should be pointed out that obtaining a better understanding of the remediation mechanism involved in S/S processing will contribute to facilitating technical improvement, further extension, and application.

## 1. Introduction

In the past few decades, the scale of the contaminated sites has increased in China due to the rapid development of industrialization and urbanization. To stop the further extension of soil pollution, a variety of environmental policies have been implemented since the initial 12th Five Year Plan. Moreover, the financial input was estimated to be increased to CNY trillions in 2020 to encourage technological innovation and improvement in the remediation of contaminated sites [1]. On this basis, quite a few technologies are proposed and subsequently proven to be available to treat heavy metal-contaminated soils, such as physical separation, incineration, solidification/stabilization (S/S), thermal desorption, and so on [2,3,4,5]. Among the various innovative technologies, the S/S method has been extensively adopted and investigated. For instance, the *Annual Report on Site Remediation Technologies* (ASR, the 12th edition) published that the S/S method was used in 217 of the 977 sites that were financially supported by the U.S. Superfund between 1982 and 2005. In China, S/S has also been increasingly used in the remediation projects of heavy metal contaminated sites. This has been the case due to its advantages of effective contaminant immobilization, significant engineering improvements, high heat and biodegradation resistances, notable economic effects, and construction convenience during in situ or ex situ remediation [6,7,8,9,10].

After being treated with S/S technology, the heavy metal ions can be effectively immobilized, while the engineering properties of the contaminated soil can also be improved [11,12,13,14]. It can be concluded that remediation efficiency predominantly depends on the interactions between binders and contaminant-bearing wastes, which involve the formation of monolithic solids with low permeability to prevent contaminant migration as well as the transformation of contaminants into chemically insoluble and stable phases [15,16,17,18]. Thus, choosing an excellent binder can largely ensure the behavior of treated soils. According to previous studies, cementitious materials, such as cement, lime, and fly ash, play important roles in the S/S treatment of contaminated soils due to their hydraulicity and self-hardening characterizations [19,20,21,22,23]. When considering the presence of heavy metal ions, the S/S process involved in cementitious materials is very likely to be altered, leading to a different performance. This is because heavy metal ions have obvious impacts on the cement hydration process, such as retarding the early hydration process, promoting the long-term hydration process, and changing the crystal morphology of hydrated products [11,11,24,25,26]. On this basis, it is of great importance to obtain a greater understanding of the interactions between heavy metal ions and binders, which involve contaminant immobilization as well as disturbed cement hydration owing to the presence of contaminants.

According to previous studies, remediation of heavy metal-contaminated soil with S/S was always a case study. It is difficult to draw consistent conclusions from the existing studies, especially when taking into account the different heavy metal species, regarding the interactions between heavy metal ions and traditional binders (such as cement, lime, and fly ash). This will make it difficult to use S/S technology in the field. Thus, an overview of the S/S mechanisms based on conventional binders, the immobilization mechanisms of different heavy metal ions, and the interference impacts of these contaminants on the S/S process and the S/S performance is carried out in the present paper. It can be expected that this study will contribute to facilitating technical improvement, further extension, and application of S/S technology.

## 2. S/S Mechanisms Based on Conventional Binders

### 2.1. Cement

Cement is predominantly composed of calcium silicate, calcium aluminate, and a small quantity of gypsum, which is generally used to moderate the initial period of the hydration rate [27]. When the cement is in contact with water, the hydrated reaction starts immediately; furthermore, the aluminate phases exhibit the fastest hydration rate in the initial period of hydration [28,29,30]. In addition to the calcium aluminate hydrates (CAHs), ettringite (AFt) can also form in the early period of cement hydration, as long as a sufficient quantity of sulfate is contained in the cement paste [31,32,33]. However, the ettringite will gradually transform into calcium monosulfoaluminate hydrates (AFm) following an increase in the hydration time due to the dilute concentration of sulfate [34,35]. After 5 h of cement hydration, a considerable growth rate of calcium silicate hydrates (CSHs), which are produced by calcium silicate hydration, can be monitored. This composes approximately 50% of the cement hydration products [36,37]. In addition, calcium hydroxide (CH) is another primary product of calcium silicate hydration, the content of which is about 25% wt. [38,39,40]. The formulations of the cement paste hydrations are summarized in Table 1.

In addition to cement, lime and fly ash are both commonly used binders to treat heavy metal-contaminated soils when based on the S/S method. This is due to their self-hardening and pozzolanic characteristics [19,20,22,41].

### 2.2. Lime

The hydration of lime can significantly increase the soil pH due to the formation of Ca(OH)_2_. When the produced Ca(OH)_2_ is sufficient, the oversaturated Ca(OH)_2_ will precipitate and fill in the soil pores, which enhances the soil’s mechanical performance [42]. Meanwhile, according to the cation exchange reactions between the free Ca^2+^ and Na^+^ (K^+^) located on the surface of soil particles, the aggregation of soil particles occurs due to the decrease in the thickness of the double diffusion layer [43,44,45]. Thus, a denser structure can be achieved. Furthermore, the silica and alumina in the soil can be dissolved in a high alkaline environment (pH ≥ 12.4), which is provided by the hydration of the lime [46]. Then, the pozzolanic reactions will take place according to the interactions between the Ca(OH)_2_, pozzolans (i.e., the silicate and aluminate phases), and water, as expressed in Equations (1) and (2). The pozzolanic products, such as CSHs and CAHs, tightly bound the soil particles together, resulting in a considerable growth in soil strength [41,47,48].
SiO_2_ + xCa(OH)_2_ + (n − 1)H_2_O = xCaO·SiO_2_·nH_2_O(1)
Al_2_O_3_ + xCa(OH)_2_ + (n − 1)H_2_O = xCaO·Al_2_O_3_·nH_2_O(2)

### 2.3. Fly Ash

Fly ash is predominantly constituted of SiO_2_, Al_2_O_3_, CaO, and Fe_2_O_3_, and its self-cementing characteristic depends on the free CaO (lime) content [49,50,51]. In addition to the self-hardening, which is due to CaO hydration, the released OH^−^ can also facilitate the dissolution of the SiO_2_ and Al_2_O_3_ contained in the fly ash [52,53], subsequently leading to the formation of CSHs and CAHs [54,55,56,57], which are identified by the XRD and SEM analysis [20,58,59,60]. Furthermore, as the fly ash is applied to modify soil performance alone, its physical characteristics (such as filling voids between the soil particles, thus resulting in a denser structure) are more pronounced than its chemical characteristics. It should be noted that, in order to accelerate the pozzolanic reactions, the utilization of fly ash is commonly blended with a certain quantity of lime or cement, particularly for Class F fly ash (with a low content of free CaO) [22,61,62,63].

As mentioned above, the main interactions that are responsible for the improvement of soil behavior between the three types of binders and soil can be summarized as follows: (1) gelification, which is attributed to the formation of CSHs and CAHs; (2) the hardening that results from Ca(OH)_2_; and (3) the aggregation of soil particles, which is owing to cation exchange. However, during the modification process, the most outstanding interaction varies based on the type of binder. As shown in Figure 1, the primary solidification mechanisms of cement and lime are the gelification of CSHs/CAHs and the hardening of Ca(OH)_2_, respectively, while fly ash can work as a filler to enhance the compaction characteristics of soil due to its fine particle size, which is more pronounced than its cementitious performance. In addition, the interactions between the soil particles and binders can be directly observed based on SEM, as shown in Figure 2.

## 3. Immobilization Mechanisms of Heavy Metal Ions in the S/S Matrix

During the S/S process, the immobilization of heavy metal ions is dependent on the development of hydrated and pozzolanic reactions. Essentially, the immobilization mechanisms of heavy metal ions based on cementitious binders can be summarized as physical encapsulation, precipitation, incorporation, and substitution, as shown in Figure 3. However, the governed immobilization mechanism seems to be different for each heavy metal ion.

### 3.1. Lead

Precipitation as insoluble lead silicates (such as PbSiO_3_, Pb_2_SiO_4_, Pb_3_SiO_5_, and Pb_4_SiO_6_) is the predominant immobilization mechanism of Pb^2+^ in cementitious materials [64,65,66,67,68]. Additionally, in the alkaline environment that is offered by cement hydration, Pb^2+^ is ready to precipitate as a lead hydroxide and lead carbonate, or occasionally as a lead sulfate, a lead hydroxyl carbonate (PbSO_4_, 3PbCO_3_·2Pb(OH)_2_·H_2_O), and a lead carbonate sulfate hydroxide (Pb_4_SO_4_(CO_3_)_2_(OH)_2_) [10,69]. According to the logarithm of the stability constants of the formation of lead hydroxide and carbonate (10.9 and 13.1, respectively), lead carbonate appears to be more preferentially formed than that of hydroxide [70,71] solids. However, compared with the lead silicates, the other lead compounds are unstable when exposed to a highly alkaline environment [72]. This is especially the case for lead hydroxide due to the fact that it will decompose as the pH increases up to 12, thus leading to the formation of a lead complex [69]. Li et al. [1] reported that this lead complex can be incorporated into a C-S-H structure. C-S-H phases have layered crystal structures that consist of the alternative tetrahedral Si-O and octahedral Ca-O chains, as well as certain water molecules and cations (in order to balance the negative charges of the structure) in the interlayer spaces [73,74,75]. Thus, during C_3_S hydration, the hydrolytic lead complex can be directly linked onto the end of the tetrahedral Si-O chain in the form of a tetrahedral Pb-O. Note that the Pb-O tetrahedron and Si-O tetrahedron are connected by one shared oxygen atom [75]. In addition, Guo et al. [76] obtained a similar result, where pozzolanic components, such as SiO_2_ and Al_2_O_3_, would dissolve into [SiO_2_(OH)_2_]^2−^ and [Al(OH)_4_]^−^ during pozzolanic reactions that can coordinate with the dissolved [Pb(OH)_3_]^−^ and [Pb(OH)_4_]^2−^ that are presented in the high alkaline environment, in which Pb^2+^ is immobilized by forming Pb-O-Si, Pb-O-Al, and Si-O-Pb-O-Al bonds. All the above-mentioned immobilization forms of lead were simultaneously detected by X-ray Absorption Fine Structure spectroscopy and Raman spectroscopy, which were performed by Contessi et al. [16]. While incorporation into the C-S-H structure is the most pronounced compared to the other lead immobilization mechanisms.

In addition to the aforementioned immobilization mechanisms, some other mechanisms were proven to be available for the lead immobilization that occurred during the S/S process. Chen et al. [77] found that Pb^2+^ immobilization was significantly impacted by the Pb^2+^ concentration: at a lower concentration, Pb^2+^ was fixed in the C-S-H by physical and chemical adsorption; at a high concentration, Pb^2+^ was immobilized by precipitation. In addition to the calcium silicate phases, certain studies have reported that the calcium aluminate phases can also make some contribution toward reducing Pb^2+^ mobility by substituting for Ca^2+^ in the hydrated aluminate phases [20,78,79]. Furthermore, Pb^2+^ can also be encapsulated by cementitious hydrates [11].

To sum up, precipitating hydroxide, silicates, and carbonate and incorporating them into a C-S-H structure by forming a Si-O-Pb bond are the most common results with respect to the lead immobilization mechanism.

### 3.2. Zinc

Numerous other previous studies have suggested that precipitation as hydroxides or carbonates may be the major mechanism of Zn^2+^ immobilization in the S/S system [64,80,81]. Zinc is a typically amphoteric metal that makes it soluble in a high alkaline environment, which is provided by cement hydration, and exists in the form of a hydroxyl complex, i.e., [Zn(OH)_4_]^2−^ and [Zn(OH)_3_]^−^ [82]. These hydroxyl complexes are hardly able to adsorb onto the surface of C-S-H due to their negative charge. Poon et al. [83,84] tested this conclusion by performing a leaching test on the Zn-bearing cement pastes, which showed that there was no Zn^2+^ leaching out during the hydrolysis of CSH. Du et al. [21] validated the absence of Ca(OH)_2_ in the hydrated Zn-bearing cement pastes based on XRD and SEM analysis, which proved that Zn^2+^ immobilization was associated with Ca(OH)_2_. A calcium zincate hydrate (CaZn_2_(OH)_6_·2H_2_O), other than C-S-H and Ca(OH)_2_, precipitated at the interface between the tri-calcium silicate paste (C_3_S) and the zinc wire, as observed by Tashiro and Tatibano [85] and Lo et al. [86]. Additionally, Liu et al. [7] also observed the presence of both zinc hydroxide and calcium zincate hydrate, which were responsible for the zinc immobilization at a high level of zinc concentration. Mollah et al. [87] made a further verification of the formation of this precipitate by Fourier transformed infrared (FTIR) spectroscopy. Yousuf et al. [88] revealed the precipitation mechanism of the calcium zincate hydrates based on the charge diffusion model. The model illustrated that the negative charges carried by the C-S-H surface constituted the first diffusion layer. In order to balance the charge, the free Ca^2+^ contained in the pore solution migrated toward the C-S-H surface and formed the second diffusion layer. Under a strongly alkaline condition, the Zn(OH)_2_ decomposed into [Zn(OH)_3_]^−^ or [Zn(OH)_4_]^2−^, which constituted the third diffusion layer. During the diffusion process, the hydroxyl complex had little interaction with the C-S-H surface; however, it reacted with Ca^2+^, which resulted in the precipitation of a calcium zincate hydrate [14,89].

Having said this, certain researchers have advised that calcium zincate hydrate is a transitional product that will decompose with the development of cement hydration, which can then eventually be incorporated into the C-S-H structure. Ziegler et al. [90] proposed that Zn^2+^ can be incorporated into the C-S-H structure by forming a Si-O-Zn bond. Liu et al. [91] reported that CSH containing Zn^2+^ was detected in S/S-treated soil by SEM testing equipped with EDS. It is speculated that Zn^2+^ is immobilized by incorporation into the interlayer of CSH under a lower Zn^2+^ concentration condition. Rose et al. [92] illustrated this binding process based on EXAFS and 29NMR analysis, which revealed that a Zn-O tetrahedron can coordinate directly with the tetrahedral Si-O chain by a shared oxygen atom. Meanwhile, several studies have suggested that the mechanisms controlled by Zn^2+^ immobilization are closely associated with the initial Zn^2+^ concentration and the pH levels of the binding system [90,93]: at a lower Zn^2+^ concentration (<1 mmol/L), chemical incorporation into C-S-H served as the major mechanism, which accounts for the Zn^2+^ immobilization when the pH varies from 11.7 to 12.8; at a higher Zn^2+^ concentration, Zn^2+^ is predominantly immobilized by precipitation as β-Zn(OH)_2_ at pH < 12, or when forming CaZn_2_(OH)_6_·2H_2_O at 12.5 > pH > 12.

Komarneni et al. [94] indicated that a small amount of Zn^2+^ can substitute for Ca^2+^ or for the Na^+^ that is located on the surface of C-S-H, as well as for the Na^+^ that exists in the interlayer of the C-S-H structure. However, Rose et al. [92] speculated that it was difficult, in terms of geometry, for Zn^2+^ to replace Ca^2+^ in the C-S-H structure. This is because Zn^2+^ and Ca^2+^ are located in the center of the Zn-O tetrahedron and Ca-O octahedron, respectively, which makes it difficult to complete the substitution. Zn^2+^, however, can take the place of Ca^2+^ in either the poorly crystalline or amorphous C-S-H.

In addition, Poon et al. [83] also found that the hydrolysis of AFt was accompanied by the apparent release of Zn^2+^, which indirectly confirmed that AFt may also be responsible for Zn^2+^ immobilization. Meanwhile, Kumarathasan et al. [95] verified that Zn^2+^ can replace Ca^2+^ in the ettringite structure.

### 3.3. Chromium

Chromium has two oxidation states, Cr^3+^ and Cr^6+^, in the natural environment. Cr^6+^ appears to be more toxic and mobile than Cr^3+^, which makes it more difficult to fix [96,97]. Based on the review of the literature, both Cr^3+^ and Cr^6+^ immobilization are closely associated with the hydration of the aluminate phases contained in the cement.

A series of analogous calcium aluminate hydrates (CAHs) (Ca_2_Cr(OH)_7_·3H_2_O were generated at 55 °C, and Ca_2_Cr_2_O_5_·6H_2_O was generated at 25 °C) bearing Cr^3+^ were detected by Kindness et al. [98] when Cr^3+^ reacted with Ca(OH)_2_ in the presence of water. For the sake of affirming this fixation mechanism of Cr^3+^, Kindness et al. [98] proposed another test program in which the generation of Ca_2_Cr_2_O_5_·6H_2_O based on the interactions between Cr^3+^ and pure tri-calcium aluminate was recorded. Thus, it was concluded that Cr^3+^ was immobilized by substituting for Al^3+^ in the crystal structure of CAHs. Similar results were obtained by Leisinger et al. [99], Zhang et al. [100], and Sophia et al. [101], who observed that Cr^3+^ was fixed in two forms of Ca-Cr compounds: one was ettringite analogues and the other was monosulfoaluminate analogues, which formed in the early and later periods of cement hydration, respectively.

Jing et al. [102] suggested that the Cr^3+^ in the hydrated aluminates was only available at a pH level of no less than 10.5. However, under a relatively low alkaline condition, the precipitation of chromium hydroxides may be the major immobilization mechanism [59,103]. Moreover, Cocke [104] and Ecke et al. [105] emphasized that Cr(OH)_3_ only existed in an insoluble form in a weakly alkaline environment and was intended to decompose into Cr(OH)_4_^−^ as the pH increased to a certain high value.

In addition, based on the Richardson and Groves model [106], Wang et al. [107] and Heimann et al. [108] proposed that the replacement of Si^4+^ in the C-S-H structure by Cr^3+^ occurred during cement hydration, in which the monovalent cations (Na^+^ or K^+^) compensated for the charge deficiency. However, Omotoso et al. [109] suggested that such a substitution can hardly take place for Cr^3+^ but rather that incorporation into the C-S-H structure by a coordination with a Ca-O octahedron and Si-O tetrahedron simultaneously in the form of a Cr-O octahedron [110].

For Cr^6+^, it is difficult to immobilize by precipitation or substitution for Al^3+^ in the aluminate phases. Several researchers have suggested reducing Cr^6+^ to Cr^3+^ first with certain reductants [111,112], such as Fe^2+^, and then performing the next stabilization/solidification process. However, Dermatas and Meng [58] found that Cr^6+^ can hardly be reduced to Cr^3+^ by Fe^2+^ under alkaline conditions, which is in favor of being encapsulated by the products of pozzolanic reactions. In the alkali-activated fly ash system, Na_2_CrO_4_·4H_2_O was responsible for the Cr^6+^ immobilization, although it has a relatively high solubility [54,65]. Muhammad et al. [113] advised that CrO_4_^2−^ can react with Ca^2+^ and water under a high alkaline environment (pH > 12) provided by cement hydration to form CaCr(OH)_4_·H_2_O and CaCrO_4_·2H_2_O. Nevertheless, the most commonly accepted mechanism for Cr^6+^ immobilization is that Cr^6+^ appears as CrO_4_^2−^ and can also be incorporated into the AFt structure by the isomorphous substitution of SO_4_^2−^ contained in AFt [97,114,115].

It is interesting that the immobilization of both chromium oxidation states is predominantly attributed to the formation of analogous aluminate hydrates containing chromium.

### 3.4. Arsenic

In nature, arsenic also exists in two oxidation states, As^3+^ and As^5+^. In the S/S process, the immobilization mechanisms of the two species are quite different: As^3+^ is immobilized by forming insoluble calcium arsenate precipitates (Ca-As-O, CaAs_2_O_6_, Ca_2_As_2_O_7_, CaO-As_2_O_5_, and Ca_3_As_2_O_8_), calcium hydrogen arsenate (CaHAsO_3_, CaHAsO_4_, and Ca(H_2_AsO4)_2_), or calcium arsenate hydrates (Ca_5_H_2_(AsO_4_)_4_·9H_2_O, CaHAsO_4_∙2H_2_O, Ca_5_H_2_(AsO_4_)_4_∙5H_2_O, and CaAsO_3_(OH)∙2H_2_O) [116,117]; while As^5+^ is incorporated into NaCaAsO_4_∙7.5H_2_O whose generation is strongly dependent on the contents of Na^+^ and Ca^2+^ contained in the binder systems [118,119]. In addition, some research revealed that As immobilization was not only up to the As species but also highly associated with the molar mass ratio of Ca to As, as shown in Table 2 [119,120,121].

In addition to the aforementioned conclusions, several different results were reported by other researchers. Mollah et al. [122] claimed that replacement of the aluminate in etteringite by [AsO_4_]^3−^ can only account for the short-term As^5+^ immobilization in a cement matrix. While the formation of NaCaAsO_4_·7.5H_2_O and Ca_5_(AsO_4_)_3_OH were responsible for the long-term stabilization of As^5+^, which were identified by XRD in 10 year-old cement pastes [123]. Beiyuan et al. [117] identified that As can be strongly immobilized by the iron oxides/hydroxides and CSHs by adsorption and/or co-precipitation. Li et al. [116] stated that the potential mobility and exposure pathway of As were reduced by means of physical encapsulation in the S/S matrix, where an interlocking framework of hydration products would increase with cement content.

### 3.5. Cadmium

In cement hydration systems, the major immobilized mechanism of Cd^2+^ is precipitation of hydroxides [20,107]. Subsequently, these insoluble phases were adsorbed onto the C-S-H surface or through filling in the pore structure of the cement pastes [124,125]. Hale et al. [126] and Pandey et al. [127] advised that the precipitation of Cd(OH)_2_ was the predominant form of immobilization, which was then encapsulated by CSHs at a microscopic scale. In contrast, Conner and Reinhold [128] and Park [129] proposed that Cd^2+^ can also be immobilized by forming a double cadmium calcium hydroxide compound (CdCa(OH)_4_). Additionally, Mollah et al. [130] explained the formation of CdCa(OH)_4_ via the interfacial interactions between [Cd(OH)_4_]^2−^ and Ca^2+^, which was very similar to the formation mechanism of CaZn_2_(OH)_6_·2H_2_O [87].

Except for the precipitation of cadmium hydroxides, the other mechanisms accounting for cadmium immobilization are rarely mentioned.

### 3.6. Nickel and Cobalt

In cementitious binder systems, Ni^2+^ is commonly immobilized by the precipitation of hydroxides [131,132]. Vespa et al. [133] detected the layered Ni-Al hydroxide compounds (Ni_2_Al(OH)_6_(CO_3_)_1/2_) in addition to α-Ni(OH)_2_ and β-Ni(OH)_2_ by the EXAFS (Extended X-ray Adsorption Fine Structure) technology. Additionally, Scheidegger et al. [134,135] illustrated the precipitation mechanisms of this Ni-Al hydroxide compound with Equation (3):(3)$${\left[M_{1-x}^{2+}M_{x}^{3+}{\left(OH\right)}_{2}\right]}^{x+}(x/n)A^{n-}\cdot mH_{2}O,$$
where M^2+^ and M^3+^ presented the divalent and trivalent cations, such as Mg^2+^, Ni^2+^, Co^2+^, Zn^2+^, Mn^2+^, Fe^2+^, Al^3+^, Fe^3+^, and Cr^3+^, A^n−^ presented the monovalent or divalent anions, such as Cl^−^, ${NO}_{3}^{-}$, ${CO}_{3}^{2-}$, and ${SO}_{4}^{2-}$. Moreover, Vespa et al. [133] observed that α-Ni(OH)_2_ would gradually transform into Ni_2_Al(OH)_6_(CO_3_)_1/2_ with the process of cement hydration, while the β-Ni(OH)_2_ content remained approximately constant.

Vespa et al. [136] observed that Co^2+^ predominantly precipitated as Co(OH)_2_ during the initial hydration period. In a highly alkaline environment (pH > 12.5) or in the presence of O_2_, Co^2+^ was readily oxidized to Co^3+^ and eventually precipitated as an insoluble CoOOH. However, Hale et al. [126] stated that Co^2+^ can hardly be oxidized and that Co^3+^ can also be reduced to Co^2+^ in the presence of Fe^2+^ or Al^3+^. Catalano et al. [137] indicated that Co^2+^ can react with aluminate phases under strongly alkaline conditions that result in the formation of Co(OH)_2_, CoCO_3_, and Co_0.75_-Al_0.25_(OH)_2_CO_3−x_H_2_O.

In addition, Komarneni et al. [94] found that Ni^2+^ and Co^2+^ can replace Ca^2+^ in C_3_S or C-S-H by cation exchange reactions. Depending on the leaching test performed on the Ni^2+^/Co^2+^ bearing C_3_S/C-S-H, a considerable amount of Ca^2+^ leaches out, and simultaneously, the Ni^2+^/Co^2+^ concentration in the pore solution decreases significantly.

### 3.7. Mercury

The immobilization mechanism with respect to mercury is particularly different from the other waste ions. In that there are no chemical interactions between the Hg^2+^ and cement pastes, and there is only a physical encapsulation of the cement hydrates [138,139]. Cocke [104] stated that mercury can exist in the form of insoluble mercury oxide under alkaline conditions. Donatello et al. [140] studied Hg immobilization in alkali-activated fly ash cement. Results showed that precipitates that appeared to be yellow/orange could be occasionally encountered in the Hg-doped specimens. The SEM-EDX results proved that these precipitates were composed of HgS, Hg_2_S, and HgO. Furthermore, the leaching test ensured that the precipitation of amorphous HgS or Hg_2_S was partially accountable for Hg immobilization in Hg-doped pastes. Ortego et al. [141] stated that during cement hydration, HgO, when in a yellow form, was predominantly responsible for Hg immobilization at room temperature. However, the HgO was still exposed to an emission risk when the temperature was elevated.

Table 3 summarizes the immobilization mechanisms for the different heavy metal ions in the S/S system, and the technologies that can be used to distinguish each immobilization mechanism are listed in Table 4. As shown in Table 3, these immobilization mechanisms can be divided into four categories: (1) the precipitation of insoluble compounds; (2) the incorporation into CSH/CAH structures; (3) substitution; and (4) physical encapsulation by cementitious products. Currently, precipitation, incorporation, and substitution have been commonly accepted as the predominant immobilization mechanisms for heavy metal ions. Furthermore, they have been directly verified by some microtesting techniques. The view is that immobilization via replacement of Ca^2+^/Si^4+^ in cementitious products and in physical encapsulation remains controversial. Furthermore, it is proposed to be dependent on indirect results. Finally, it should be noted that the ultimate mechanism that is responsible for the immobilization of heavy metal ions is closely associated with the species, valence, and initial concentration of the heavy metal ions, as well as the pH level of the S/S system and hydration times.

## 4. Interference Impacts of Heavy Metal Ions

Following the hydrated and pozzolanic reactions, the heavy metal ions were immobilized by the corresponding products. Simultaneously, the presence of heavy metal ions can also impose certain interferences on the above reactions. Consequently, the quantity and morphology of the cementitious products may be altered, and they may subsequently influence the interactions between soil particles and cementitious products as per the performance of the treated soil. Considering the fact that different heavy metal ions are immobilized in different ways in the S/S system, it can be speculated that different interferences can be aroused during the hydrated and pozzolanic reactions.

### 4.1. Hydrated and Pozzolanic Reactions

Based on the study of Tay [142], the standard initial setting time of OPC should not be less than 45 min, while the final setting time should be more than 10 h. However, when the heavy metal species are poured into the cement pastes, the setting time is very likely to be altered. According to previous studies [16,73,143], the interference impacts of heavy metals on cement hydration are closely associated with the heavy metal species and their corresponding concentrations in the cement hydration system.

#### 4.1.1. Lead

It is generally concluded that the presence of Pb^2+^ induces the strongest retardation of cement hydration (including both initial and final setting times) when compared with other heavy metals [144,145,146]. The retardation of cement hydration that is caused by Pb can be primarily attributed to the formation of Pb precipitates covering the area around the cement grains [104,110,147,148].

Ortego et al. [141] suggested that Pb^2+^ primarily adheres to the surface of hydrated cement as a sulfate or hydroxo-sulfate species, as determined via XPS and FTIR analysis. Thus, the cement grains are prevented from coming into contact with water, resulting in the retardation of the hydration process. The aforementioned reaction consumes part of the sulfate in the hydration system, which leads to a reduction in sulfate concentration. It is well known that during the early hydration period, the formation of ettringite requires a relatively high sulfate concentration. Therefore, the presence of Pb^2+^ would inhibit ettringite and monosulfate formation if the remaining sulfate concentration in the hydration system was not enough. Mohammed et al. [149] suggested that ettringite cannot form in the presence of Pb^2+^. This is because the sulfate in the hydration system will react first with Pb^2+^. On this basis, the influence of Pb^2+^ on the formation of ettringite is most likely determined by the Pb^2+^ concentration.

Cocke [104] and Chen et al. [77] proposed that Pb^2+^ influenced the cement hydration by decreasing the silicate polymerization, which is closely associated with the Pb^2+^ concentration. Otherwise, the retardation impact of Pb^2+^ on the cement hydration was not obvious for the specimens that were cured for 28 days but was significant for the samples that were cured for 48 h and 7 days [150].

Hills et al. [151] proved that Pb^2+^ accelerated C_3_S hydration besides aggravating the carbonation during cement hydration. As a result, the hydrated product, Ca(OH)^2^, will transform into calcite. Based on the TGA results, the portlandite content of the control cement paste is 16.7%, and the calcite content is 16.8%. While for the Pb-doped cement pastes, the calculated portlandite content is 7.1% and that of the calcite is 27.7%. This indicates that the presence of Pb^2+^ retards the precipitation of portlandite; meanwhile, it also accelerates the carbonation of portlandite due to the lower pH that is attributed to the hydrolysis of Pb^2+^.

#### 4.1.2. Zinc

With the exception of lead, zinc is another heavy metal that has severe impacts on cement hydration. The retardation of both calcium aluminate and calcium silicate phases in the cement occurs in the presence of Zn^2+^ [152,153,154]. The addition of Zn^2+^ decreases the ratio of Ca/Al in the pore solution of cement paste because of the interactions between Zn^2+^ and CaO [155]. Thus, the formation of C_3_AH_6_ is hindered. Moreover, the impact of Zn on the hydration of the C_3_A phase is closely associated with the sulfate content in the cement. The C_3_A hydration will not be retarded until the sulfate content is higher than 2.5% [156]. Based on previous studies, the retardation of calcium silicate phases can be attributed to the precipitation of insoluble zinc compounds that encapsulate the cement grains [77,152,154]. However, Malviya and Chaudhary [81], as well as Hamilton and Sammes [157], indicated that the retardation of C_3_S hydration was closely associated with Zn^2+^ concentration. As such, they concluded that no retardation could be observed at a low Zn^2+^ concentration. While the calcium zinc hydrate cannot be detected at 28 days of curing until the zinc concentration in the cement paste increases to 10%, Yousuf et al. [88] identified this compound as CaZn_2_(OH)_6_·2H_2_O via FTIR analysis, which seemed to form a protective layer that impedes the normal hydration of cement grains.

The XRD results obtained by Hills et al. [151] showed that the peak intensity of the C_3_S of Zn-doped cement pastes was much higher than that of the control paste. It was suggested that the presence of Zn^2+^ retards the hydration of C_3_S. However, this retardation effect was weakened when increasing the curing time.

#### 4.1.3. Chromium

Chromium imposes very different impacts on the cement hydration when compared with lead and zinc, which tend to accelerate the initial cement hydration or produce little interference [92,156,158].

When the Cr_2_O_3_ content is low (≤2.5%), it has no influence on the setting time [156]. Tashiro and Oba [153] also found that cement pastes that were doped with 1% and 5% Cr_2_O_3_ had similar C_3_AH_6_ formation to that of control cement paste. While at a higher Cr_2_O_3_ content, the hydrates were clearly distinguished after 28 days of curing. It can be speculated that the effect of Cr on cement hydration is predominantly dependent on its concentration.

Chen et al. [77] suggested that the degree of C_3_S hydration was accelerated by the presence of Cr^3+^ because the attack of H^+^ resulted from chromium hydrolysis. Another explanation emphasizes that chromium compounds are not formed on the surface of hardened cement but that they disperse below the surface of the OPC matrix [130]. As a result, the cement’s hydration cannot be hindered by insulating the cement grains from the water. However, Li et al. [159] reported that Cr^3+^ accelerated the hydration of C_3_A at 3 days but restricted it at 7 days. An analogous result was obtained by Rha et al. [160], who pointed out that the presence of Cr^3+^ could accelerate the initial hydration of slag-cement but lead to a reduction in hydration degree. On the one hand, the presence of Cr^3+^ can restrain ettringite formation, which is considered the setting delay of calcium silicates in the initial period of hydration. On the other hand, Cr^3+^ can interact with H_2_O based on the complexation detailed as follows [161,162]:(4)$${\mathrm{Cr}\left(H_{2}O\right)}_{6}^{3+}\;\to \;\mathrm{Cr}(\mathrm{OH}){\left(H_{2}O\right)}_{5}^{2+}\;\to \;\mathrm{Cr}(\mathrm{OH}{)}_{3}(H_{2}O{)}_{3}\;\to \;\mathrm{Cr}{\left(\mathrm{OH}\right)}_{4}{(H_{2}O)}_{2}^{2-}.$$

As a result, the OH^−^ concentration in the pore solution decreases, which will weaken the hydration degree.

#### 4.1.4. Cadmium

Bobrowski et al. [163] stated that Cd^2+^ could shorten the initial setting time of cement hydration. Additionally, according to the results published by Díez et al. [164], it was found that the precipitation of Cd(OH)_2_ gave rise to the hardening of cement pastes, which accelerated the initial setting of the cement hydration. However, the formation of Cd(OH)_2_ may inhibit further hydration of the cement. According to the hydration exothermic curves obtained by Wang et al. [165], increasing the CdO dosage could result in the gradual extension of the introduction period of cement paste, followed by an increase in time spans for both the acceleration and deceleration periods. Meanwhile, the highest exothermic peaks were reduced. Previous studies have suggested that three-dimensional structures with a thickness of 0.1–0.3 μm containing heavy metal can form coatings around cement grains [166]. If these coatings were in the least insoluble phase, the hydration of encapsulated cement grains could be retarded [144,167]. Given that the solubility of Cd(OH)_2_ at 20 °C is only 2.697 × 10^−4^ g/100 g, which is considerably lower than Ca(OH)_2_ (0.173 g/100 g), Cd^2+^ could inhibit further hydration of cement by precipitating Cd(OH)_2_ around the clinker particles surface. Moreover, the XRD and SEM results obtained by Tumidajski and Thomson [168] can also confirm this conclusion, who presented findings showing that the presence of Cd^2+^ inhibited the formation of C_3_AH_6_.

#### 4.1.5. Mercury

When compared with the other heavy metals, mercury (whether as an ionic species or as a covalent molecule) appears to demonstrate quite different impacts on the hydration of cementitious materials. HgCl_2_ has little influence on the cement’s hydration, while Hg(NO_3_)_2_ retards the hydration in terms of strength development. The formation of a waterproof coating or consuming some OH^−^ by Hg^2+^ may be responsible for such retardation [167].

### 4.2. Development of Compressive Strength

Considering the impacts of heavy metals on cement hydration, the mechanical properties of the cement doped with contaminants should be distinguished from those of the pure cement paste. According to the previous studies, as shown in Table 5, it can be found that the effects of heavy metals on the strength development of cement-based materials are complex and are primarily associated with the heavy metal species, heavy metal concentrations, and curing time.

Tashiro et al. [173] reported that the presence of heavy metal oxides (ZnO, Fe_2_O_3_, Cr_2_O_3_, Cu(OH)_2_, and Pb_2_O(OH)_2_) induced an apparently reduced compressive strength of cement mortars. It is because these heavy metal oxides promote the growth of ettringite, which causes an expansion of the cement mortars. Thus, the strength is reduced due to the increase in the total pore volume. The results obtained by Guo and Shi [54] showed that the addition of 0.025% Cr^6+^ and Pb^2+^ to the class C fly ash geopolymer resulted in strength reductions of 19.4% and 15.1%, respectively. This can be predominantly attributed to the formations of Na_2_CrO_4_·4H_2_O and Pb_3_SiO_5_, which most likely have lower strengths than those found in typical cement-hydrated products. Akhter et al. [150] and Rha et al. [160] also found a strength reduction in cement mortars in the presence of Pb^2+^ due to serious retardation of the initial cement hydration. For the specimens doped with Cr^3+^, strength development was still hindered, although the presence of Cr^3+^ had little influence on the reaction rate of cement hydration [160]. Gervais and Ouki [158] investigated the effects of heavy metal ions on the strength of cement mortars. Cement mortar was blended with silica fume and zeolite, as shown in Figure 4. The results indicated that different heavy metal ions have different impacts on the strength development of the tested mortars. Pandey et al. [127] reported that the presence of heavy metals decreased the compressive strength of cement mortars by 10–31%. The order of the impacts involved in different heavy metal ions on the strength of the sample was: Cd ≤ Cr(III) ≤ Zn ≤ Pb ~ Cr(VI) ≤ Cu. While Navarro-Blasco et al. [174] obtained a different result in the study of the strength properties of aluminate cement doped with heavy metals, the results showed that cement doped with Cu obtained the maximum strength, and that of Zn was the lowest.

The variations in heavy metal concentration in the cement-based materials have an obvious influence on their strength development. Li et al. [25] and Li and Yi [175] studied the effect of Zn^2+^, Cd^2+^, and Ni^2+^ on the strength of cement soil and indicated that the strength decreased with an increase in heavy metal concentration, as presented in Figure 5. Du et al. [21] found that increasing the Zn^2+^ concentration in the cement stabilized kaolin and caused a significant reduction in strength. This is because the precipitation of CaZn_2_(OH)_6_·2H_2_O would encapsulate cement grains, which retards cement hydration [86]. While some other researchers proposed that the addition of Zn^2+^ and Pb^2+^ at a low level can enhance the mechanical performance of the cement-based materials, as the heavy metal concentration exceeded a critical concentration, the presence of heavy metals would impose a detrimental influence on the strength development of the cement-based materials [25,175,176,177]. Under a low concentration condition, lead and zinc salts intend to precipitate immediately under an alkaline environment provided by the cement hydration. These precipitates can work as the skeleton and fill in the pores of the cement paste, which has little influence on strength development or even the promotion of strength. While at a high level of concentration, total porosity increased and flexural strength decreased [173,178]. Cho et al. [179] suggested that changes in the volume of micropores with a radius less than 5 nm had little influence on the strength development of cement, but pores with a radius larger than 5 nm could significantly influence the strength. Chen et al. [24] thought that the reduction in strength was due to the addition of heavy metal at high concentration, which could be attributed to the changes in the structure (crystallinity and particle size) and solubility of the hydrated products. This is because the heavy metal ions can be adsorbed on the surface of cement grains during their hydration and then enter the lattice to form a solid solution. Moreover, Alford et al. [180] indicated that Pb-bearing precipitates formed rapidly as soon as excess Pb(NO_3_)_2_ was added to cement pastes. Subsequently, these precipitates filled in the pores along with the volume expansion, which led to microcrack formation and a resulting reduction in strength.

In addition, several studies revealed that the presence of Cr could improve the strength properties of cement pastes even though the Cr concentration was at a very high level [156,173,181]. This was probably due to the fact that Cr is incorporated into the lattice of cement hydrates during cement hydration, forming a stable structure that can induce an increase in strength [65].

Despite the presence of heavy metals having interference effects on the strength development of cement pastes, this effect will diminish with an increase in the curing time [141,153,157]. Olmo et al. [156] showed that the addition of 5% ZnO decreased the UCS at 3 and 7 days, but it obtained a similar strength to pure cement pastes as the curing time increased up to 28 days. Based on thermogravimetric analysis (TGA) and X-ray diffraction analysis, Chen et al. [77] illustrated that the presence of heavy metal ions would inhibit the formation of portlandite at a short curing period. However, increasing the curing time will weaken this inhibition’s impact.

## 5. Advantages and Disadvantages of S/S Technology

Currently, the most commonly used technologies to treat heavy metal-contaminated sites include solidification/stabilization, electrokinetic technology, soil washing, ceramic solidification, phytoremediation, etc. Compared to other technologies, S/S is a mature construction technology characterized by its excellent effectiveness in remediation, low cost, and short duration. The cost and duration of each technology are listed in Table 6.

In addition to the above-mentioned advantages, some disadvantages remain outstanding for S/S technology. The commonly used binders are almost silicate materials, the hardening paste of which has poor durability when exposed to sulfate attack. Acid rain is extensively distributed in China, which puts stabilized contaminants at risk of refiltration.

## 6. Conclusions and Prospects

During the rapid development of industrialization and urbanization, S/S technology has been increasingly used in remediation projects for heavy metal-contaminated sites. During S/S processing, the main interactions that are responsible for improving the soil’s behaviors can be summarized as precipitation, chemical incorporation, isomorphous substitution, physical adsorption, and encapsulation. Currently, precipitation, incorporation, and substitution have been commonly accepted as the predominant immobilization mechanisms for heavy metal ions and have been directly verified by some microtesting techniques. While replacement of Ca^2+^/Si^4+^ in the cementitious products and physical encapsulation remain controversial, which is proposed dependent on the indirect results. Lead and zinc can retard both the initial and final setting times of cement hydration, while chromium can accelerate the initial cement hydration. Though cadmium can shorten the initial setting time, further cement hydration will be inhibited. While for mercury, the interference impact is closely associated with its adapted anion. The complex environment of the S/S system (such as the contaminant concentration, pH level, and molar ratio of Ca/Si and Ca/Al) is noted, which makes it difficult to identify the controlling immobilization mechanism for a specific contaminant. In general, S/S-treated heavy metal-contaminated soil with conventional binders appears to have poor durability when exposed to a complicated and changeable environment, such as drying-wetting cycles, freezing-thawing cycles, acid rain leaching conditions, chloride, and sulfate attacks. In addition, the manufacture of conventional binders is always accompanied by the emission of greenhouse gases, which is actually opposite to the environmental protection policy in China. Thus, there is a pressing need to research and develop novel binders that can replace the conventional ones and have better performance. In addition, obtaining a better understanding of the remediation mechanisms involved in S/S processing is of great importance to facilitate technical improvements and further extend and apply this technology.

## Figures and Tables

**Figure 1 materials-16-03444-f001:**
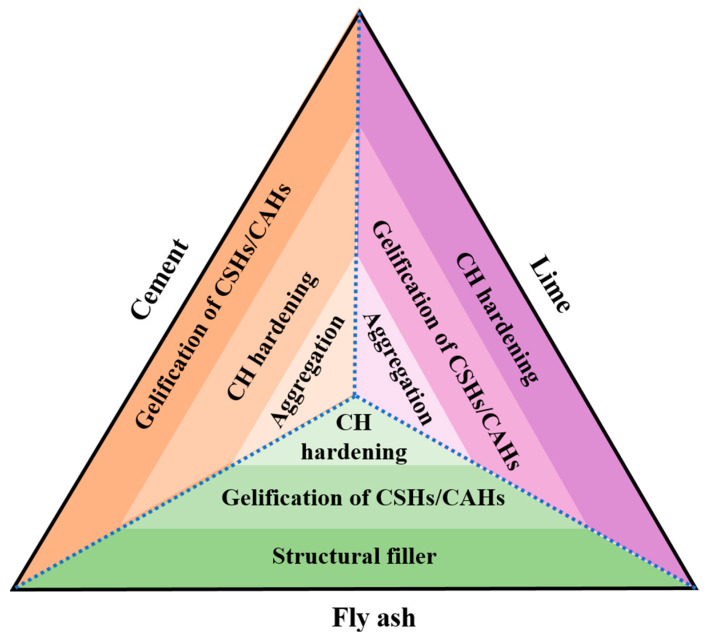
Primary solidified mechanisms of different binders during the S/S process.

**Figure 2 materials-16-03444-f002:**
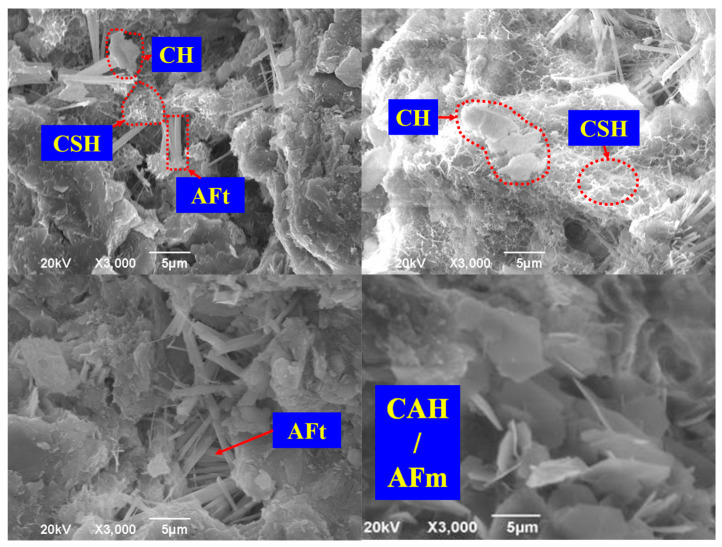
SEM images of the microstructure of the S/S-treated soils based on the conventional binders [23,57].

**Figure 3 materials-16-03444-f003:**
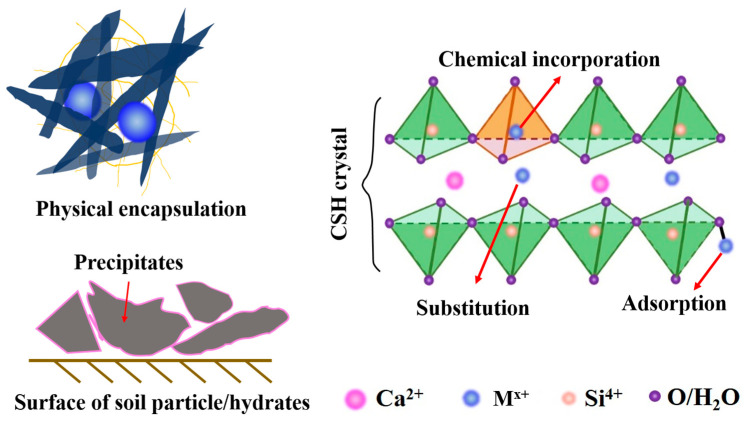
Predominant immobilization mechanisms of heavy metal ions in the S/S matrix.

**Figure 4 materials-16-03444-f004:**
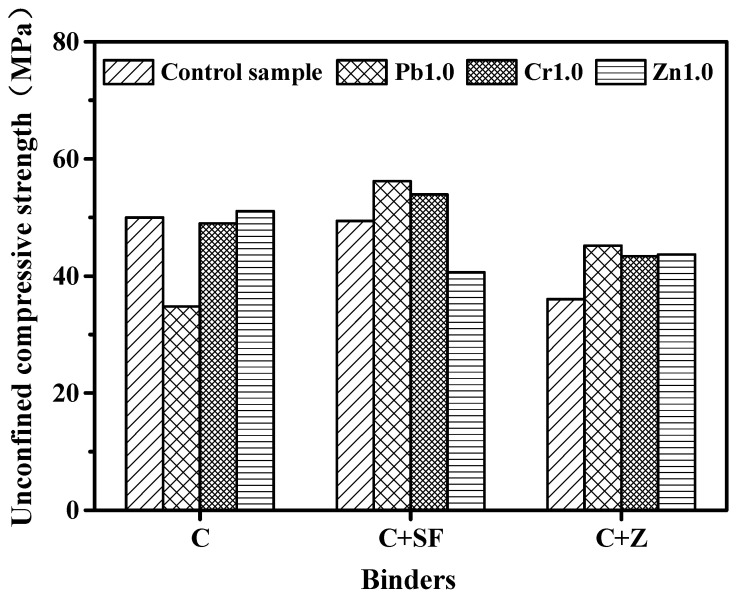
Impacts of heavy metals on the strength of cement pastes.

**Figure 5 materials-16-03444-f005:**
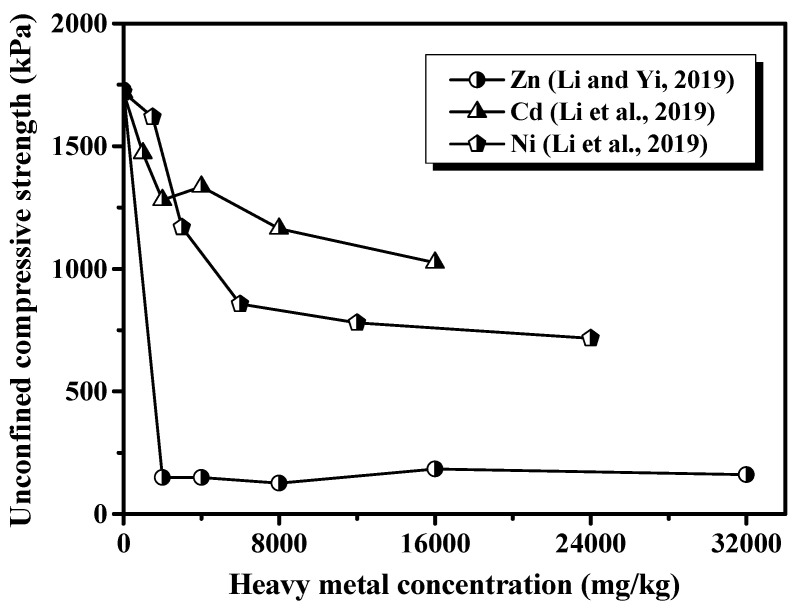
Effects of heavy metal ions on the strength development of cement soils [25,175].

**Table 1 materials-16-03444-t001:** Hydration formulations of cement pastes.

Compositions	Hydrated Reactions
C_3_S	C_3_S + H_2_O = C-S-H + Ca(OH)_2_
C_2_S	C_2_S + H_2_O = C-S-H + Ca(OH)_2_
C_3_A	C_3_A + H_2_O = C-A-H
	C_3_A + Ca(OH)_2_ + H_2_O = C-A-H
	C_3_A + 3CaSO_4_ + 32H_2_O = 3CaO·Al_2_O_3_·3CaSO_4_·32H_2_O
	C_3_A + CaSO_4_ + 12H_2_O = 3CaO·Al_2_O_3_·CaSO_4_·12H_2_O
C_4_AF	C_4_AF + H_2_O = C-A-F-H

**Table 2 materials-16-03444-t002:** Immobilization mechanisms of different As species with consideration of the molar mass ratio of Ca to As.

Species	Molar Mass Ratio of Ca:As	Mechanisms
As_2_O_3_	/	Ca-As-O
NaAsO_2_	/	Ca-As-O
Na_2_HAsO_4_·7H_2_O	Ca:As > 1.5:1	NaCaAsO_4_·7.5H_2_O
	Ca:As = 1:1; Ca:As = 1.7:1–1.9:1	Ca_5_(AsO_4_)_3_OH
	Ca:As > 1:1; Ca:As = 2:1–2.5:1	Ca_4_(OH)_2_(AsO_4_)_2_·4H_2_O

**Table 3 materials-16-03444-t003:** Immobilization mechanisms for different heavy metal ions in the S/S system.

Heavy Metal Ions	Immobilization Mechanisms		
	Primary	Secondary	Controversial
Pb(II)	Precipitation of lead silicates	Incorporation into a CSH/CAH structure	Ca^2+^ in CAHs is replaced by Pb^2+^
Zn(II)	Precipitation of zinc hydroxides	Incorporation into a CSH structure	Ca^2+^ and Na^+^ on the surface of CSHs, as well as Ca^2+^ in AFt, are replaced by Zn^2+^
Cr(III)	Cr^3+^ substitutes for Al^3+^ in CAHs	Precipitation of Cr(OH)_3_	Si^4+^ in CSHs is replaced by Cr^3+^, with monovalent cations compensating for the charge deficiency
Cr(VI)	SO_4_^2−^ in AFt is substituted by Cr^6+^ in the form of CrO_4_^2−^	Physical encapsulation	CrO_4_^2−^ reacts with Ca^2+^ and OH^−^ to form CaCr(OH)_4_·H_2_O and CaCrO_4_·2H_2_O
Cd(II)	Precipitation of Cd(OH)_2_	Precipitation of CaCd(OH)_4_	/
As(III)	Precipitation of Ca-As-O	Precipitation of Ca-H-As-O and Ca-As-H	/
As(V)	Incorporation into NaCaAsO_4_·7.5H_2_O and Ca_5_(AsO_4_)_3_OH	Substitution for aluminates in AFt with AsO_4_^3−^	/
Ni(II)	Precipitation of β-Ni(OH)_2_	Forms layered Ni-Al hydroxide compounds (Ni_2_Al(OH)_6_(CO_3_)_1/2_)	Ca^2+^ in CSHs is replaced by Ni^2+^/Co^2+^
Co(II)	Precipitation of Co(OH)_2_	Forms insoluble CoOOH	
Hg(II)	Hg^2+^ is oxidized to form insoluble HgO in alkaline conditions and is encapsulated by hydrated products	Forms amorphous HgS or Hg_2_S precipitates	Precipitation of Hg(OH)_2_

**Table 4 materials-16-03444-t004:** Technologies used to determine the immobilization mechanisms of different heavy metal ions.

Immobilization Mechanism	Technology
Precipitation	X-ray diffraction, Thermogravimetric Analysis, Scanning Electron Microscope
Incorporation	Raman spectroscopy, X-ray Absorption Fine Structure (XAFS) spectroscopy, X-ray Photoelectron spectrometer
Substitution	X-ray Photoelectron spectrometer, Energy Disperse Spectroscopy
Encapsulation	By means of a comprehensive test

**Table 5 materials-16-03444-t005:** Applications of the cementitious binders to treat heavy metal ions based on S/S technology.

References	Soil Characteristics	Species and Concentration of HMs * (mg/kg)	Binder Proportions	UCS of 28 Day (MPa)
[25]	90% fine slag 10% kaolin	Pb 1000	5% CaO	0.160
			5% MgO	0.126
		Pb 16,000	5% CaO	0.160
			5% MgO	0.802
[169]	High liquid clay	Zn 5000	6% novel cement	1.087
		Zn 10,000	6% novel cement	1.032
[114]	Red clay	Pb 2000; Zn 2000	10% cement	9.33
			10% cement + 5% fly ash	9.75
			10% cement + 5% fly ash + 2% lime	14.84
[19]	Contaminated sediment	As 2047; Pb 1677	8.45% MK + 2.13% lime + 0.51% PG	27.36
			6.11% MK + 4.62% lime + 0.37% PG	31.69
			4.31% MK + 6.53% lime + 0.26% PG	32.96
			6.30% MK + 1.59% lime + 2.84% calcite + 0.38% PG	23.77
			4.90% MK + 3.70% lime + 2.21% calcite + 0.29% PG	29.89
			3.67%MK + 5.56% lime + 1.65% calcite + 0.22% PG	38.45
[170]	Sludge	PbO 36,380	60% GGBFS + 20% fly ash + 20% sludge bearing Pb	7.54
[58]	5% kaolin 95% slag	PbO 7000	10% lime	0.154
		Cr(III) 4000	25% fly ash	3.830
		Cr(VI) 4000	10% lime + 25% fly ash	6.663
[21]	Kaolin	Zn 1000	8% cement	0.117
			12% cement	0.333
			15% cement	0.387
			18% cement	0.491
[42,171]	65% gravel 29% slag 2.8% silt 3.2% clay	Cd 3000; Cu 3000; Pb 3000; Ni 3000; Zn 3000; TPH 10,000	1% cement + 4% fly ash	0.09
			1% Ca(OH)_2_ + 4% GGBFS	0.049
			0.5% cement + 4.5% GGBFS	0.106
			2% cement + 8% fly ash	0.099
			2% Ca(OH)_2_ + 8% GGBFS	0.412
			1% cement + 9% GGBFS	0.468
			4% cement + 16% fly ash	0.45
			4% Ca(OH)_2_ + 16% GGBFS	0.581
			2% cement + 18% GGBFS	0.803
[172]	Clay	Cd 132; Pb 121; Cu 287; Ni 178; As 661; Zn 44,074	15% cement	0.63
		Cd 37; Pb 37; Cu 157; Ni 253; As 82; Zn 2074		4.18

* HMs—heavy metal ions; MK—metakaolin; PG—phosphogypsum; GGBFS—Ground Granulated Blast Furnace Slag.

**Table 6 materials-16-03444-t006:** The cost and duration of each technology.

Technology	Cost (EUR)	Duration (Month/2 × 10^4^ Tons)
Solidification/stabilization	75–200	6–12 or more
Electrokinetic technology	65–195	6–12
Soil washing	140–400	6–12
Ceramic solidification	300–400	<6
Phytoremediation	50–150	>12

## Data Availability

No new data were created or analyzed in this study. Data sharing is not applicable to this article.

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
