# Peer review of "Review of the Interactions between Conventional Cementitious Materials and Heavy Metal Ions in Stabilization/Solidification Processing"

_materials, 2023, doi:10.3390/ma16093444_

Round 1

Author Response

Comments:

  1. English must be rechecked.

Authors’ reply:

Thanks for your suggestion. The writing of this manuscript has been rechecked by a paid editing service offered by MDPI, and further revised by the authors.

The follow is the certification of English-editing.

  1. Introduction part should be more precise.

Authors’ reply:

Thanks for your suggestion. The Introduction has been fully revised, which can be found in the manuscript.

  1. The quality of the Figure 1 can be improved.

Authors’ reply:

Thanks for your suggestion. The quality of Figure 1 has been improved as follows:

  1. Equation numbers are not in order.

Authors’ reply:

Thanks for your suggestion. The equation number has been corrected.

Reviewer 2 Report

The comments are attached in the word document

Author Response

Following are the comments and suggestions which need to be addressed.

  1. The importance of this study should be highlighted in the manuscript and explain the real-time application of the cementitious materials concerning a large scale.

Authors’ reply:

Thanks for your suggestion. The writing of this manuscript has been rechecked by a paid editing service offered by MDPI, and further revised by the authors.

The follow is the certification of English-editing.

  1. The abstract should be reorganized. It should be more informative. Add some more important numerical results (experimental findings in the major review) in the abstract instead of general statements.

Authors’ reply:

Thanks for your suggestion. The revision has been done as follow:

“In the past few decades, the solidification/stabilization (S/S) technology has been put forward for the purpose of improving the soil strength and inhibiting the contaminant migration in the remediation of heavy metal contaminated sites. Cement, lime and fly ash are among the most commonly and effective binders to treat contaminated soils. During S/S processing, the main interactions that responsible for improving the soil behaviors can be concluded as gelification, self-hardening and aggregation. Currently, precipitation, incorporation and substitution have been commonly accepted as the predominant immobilization mechanisms for heavy metal ions, and have been directly verified by some micro-testing techniques. While replacement Ca2+/Si4+ in the cementitious products and physical encapsulation still remains controversial, which is proposed depending on the indirectly results. Lead and zinc can retard both the initial and final setting time of cement hydration, while chromium can accelerate the initial cement hydration. Though cadmium can shorten the initial setting time, the further cement hydration will be inhibited. While for the mercury, the interference impact is closely associated to its adapted anion. It should be pointed out that obtaining a better understanding of the remediation mechanism involved in S/S processing will facility the technical improvement, further extension and application.”

  1. The introduction should be reorganized. The most important aspects related to this topic should be clearly presented to provide a proper description of the state of the art in this field.

Authors’ reply:

Thanks for your suggestion. The revision has been done as follow:

“In the past few decades, the scale of the contaminated sites increased by years in China with the rapid development of industrialization and urbanization. In order to stop the further extension of soil pollution, a variety of environmental policies have been operated since the initial of 12th Five-Year Plan. Moreover, the financial input was estimated to be increased to trillions in 2020 to encourage the technological innovation and improvement in the remediation of contaminated sites [1]. On this basis, quite a few technologies are proposed, and subsequently proved to be available to treat heavy metal contaminated soils, such as physical separation, incineration, solidification/stabilization (S/S), thermal desorption and so on [2-5]. Among the various innovative technologies, the S/S method has been extensively adopted and investigated. For instances, the Annual Report on Site Remediation Technologies (ASR, the 12th edition) published that S/S method was used in 217 of 977 sites, which were financially supported by the U.S. Superfund between 1982 and 2005. While in China, S/S has also been increasingly used in the remediation projects of heavy metal contaminated sites. This has been the case due to……”

“According to the previous studies, remediation of heavy metal contaminated soil with S/S were always case studies. Consistent conclusions referred to the interactions be-tween the conventional binders (i.e., cement, lime, and fly ash) and heavy metal ions can hardly be reserved from the existing researches, especially with consideration of heavy metal species. This will make troubles to the field application of S/S technology. Thus, an overview on the S/S mechanisms based on conventional binders, immobilization mechanisms of different heavy metal ions, and the interference impacts of these contaminants on the S/S process and the S/S performance is carried out in the present paper. It can be expected that this study will contribute to facilitate the technical improvement, further ex-tension and application of S/S technology.”

  1. The author should show the schematic representation of the basic structural mechanism (chemical structural configuration) of solidification/stabilization techniques. Please more explain with schematic structural representation.

Authors’ reply:

Thanks for your suggestion. The basic structural mechanism of S/S techniques has been explained by SEM image and chemical structural configuration as shown in Figure 2 and Figure 3.

Figure 2. SEM images of microstructure of the S/S treated soils based on the conventional binders [23, 59]

Figure 3. Predominant immobilization mechanisms of heavy metal ions in S/S matrix

  1. The authors should compare the comparisons of specific experimental data obtained with the results of the authors of other works in the solidification/stabilization method.

Authors’ reply:

Thanks for your suggestion. The relevant result obtained from the previous studies performed by the authors has been added in the manuscript, and made the comparison with the work of the other researchers. For instance:

3.1 lead

“…Additionally, in the alkaline environment that is offered by cement hydration, Pb2+ is ready to precipitate as a lead hydroxide and lead carbonate, or occasionally as a lead sulfate, a lead hydroxyl carbonate (PbSO4, 3PbCO3•2Pb(OH)2•H2O), and a lead carbonate sulfate hydroxide (Pb4SO4(CO3)2(OH)2) [10, 71]. According to the logarithm of the stability constants of the formation of lead hydroxide and carbonate (10.9 and 13.1, respectively), lead carbonate appears to be more preferentially formed than that of hydroxide [72, 73] solids…”

“.All the above mentioned immobilization forms of lead were simultaneously detected by X-Ray Absorption Fine Structure spectroscopy and Raman spectroscopy, which was performed by Contessi et al. [16]. While incorporation into C-S-H structure is the most pronounced comparing to the other lead immobilization mechanisms…”

3.2 zinc

“…A calcium zincate hydrate (CaZn2(OH)6•2H2O), other than C-S-H and Ca(OH)2, precipitated at the interface between the tri-calcium silicate paste (C3S) and the zinc wire was observed by Tashiro and Tatibano [87] and Lo et al. [88]. And, Liu et al. [7] also observed the presence of both zinc hydroxide and calcium zincate hydrate, which were responsible for the zinc immobilization at a high level of zinc concentration. Mollah et al. [89] made a further…”

“…Ziegler et al. [93] proposed that Zn2+ can be incorporated into the C-S-H structure by forming a Si-O-Zn bond. Liu et al. [94] reported that CSH containing Zn2+ was detected in a S/S treated soil by SEM test equipped with EDS. It is speculated that Zn2+ is immobilized by incorporation into the interlayer of CSH under a lower Zn2+ concentration condition. Rose et al. [95] illustrated this binding process based on EXAFS and 29NMR analysis, which…”

  1. I should recommend to add the economic assessment/cost analysis section in the present review studies. The cost analysis section should be added for the use of cementitious materials to improve the soil strength and inhibit contaminant migration.

Authors’ reply:

Thanks for your suggestion. The economic assessment/cost analysis has been added in the manuscript section 5 and Table 6.

“Currently, the most commonly used technologies to treat heavy metal contaminated sites include solidification/stabilization, electro-kinetic technology, soil washing, ceramic solidification and phytoremediation, etc. Comparing to the other technologies, S/S is a mature construction technology characterized by the excellent effectiveness of remediation, low-cost and short duration. The cost and duration for each technologies are listed in Table 6.”

Table 6. The cost and duration for each technologies

Technology

Cost (EUR)

Duration (month/2×104 tons)

Solidification/stabilization

75-200

6-12 or more

Electro-kinetic technology

65-195

6-12

Soil washing

140-400

6-12

Ceramic solidification

300-400

< 6

Phytoremediation

50-150

> 12

  1. The authors should show the immobilization mechanisms for different heavy metal ions with results (numerical data) in various methods (precipitation, chemical incorporation, isomorphous substitution, physical adsorption, and encapsulation)

Authors’ reply:

Firstly, thanks for your suggestion. I am so sorry that the immobilization mechanisms for different heavy metal ions cannot be shown with numerical data because of the copyright restrictions. But, the technologies used to determine each immobilization mechanisms for different heavy metal ions have been listed in Table 4, as follows:

Table 4 Technologies used to determine the immobilization mechanisms of different heavy metal ions

Immobilization mechanism

Technology

Precipitation

X-ray diffraction, Thermogravimetric Analysis, Scanning electron microscope

Incorporation

Raman spectroscopy, X-Ray Absorption Fine Structure spectroscopy, X-ray Photoelectron spectrometer

Substitution

X-ray Photoelectron spectrometer, Energy Disperse Spectroscopy

Encapsulation

By means of comprehensive test

  1. Add the table for the synthesis and applications of cementitious materials.

Authors’ reply:

Thanks for your suggestion. Table 5 lists the application of cementitious materials, which can be checked in the revised manuscript.

  1. The authors must delete the old references and should include the new references (After 2019) with the same concept. So that the research in this field was updated.

Authors’ reply:

Thanks for your suggestion. The recent references cited from the high-impact journals, such as Journal of hazardous materials, Chemosphere, Construction and building materials, Cement and concrete researches, Soils and Foundations, Science of the Total Environment, have been added in the manuscript, following by the delete of the old references. It can be checked in the revised manuscript.

Reviewer 3 Report

The paper “Consistent description of the interaction mechanism between conventional cementitious materials and heavy metal ions based on stabilization/solidification (S/S)” can be accepted for publication, but several areas need improvement. I recommend this manuscript for publication after the clarification of the following major comments. See the comments below:

1. Paper title needs to revise.

2. Abstract: Need to revise. In the abstract, add a description to clarify the flow of work.

3. Authors are advised to revise the abstract and please focus abstract on the review approach.

4. Some keywords should be revised in the manuscript.

5. The manuscript is in itself very hard to read. Several instances of repeated sentences have been used to describe the same thing. This has to be extensively taken care of throughout the manuscript.

6. There are multiple grammatical errors in the manuscript. The manuscript could utilize some improvement in English by checking it with a native English speaker or software which provides their services for improving the quality of the manuscript.

7. The manuscript also provides a very general overview of the topic. The review should include more critical inputs including comparisons between various literature and a critical assessment of the approach’s pros and cons.

8. The introduction section needs to be revised. Some more recent developments in this field are required to incorporate into the introduction section.

9. Figure-1 quality needs to improve.

10. The explanation of the section “3. Immobilization mechanisms of heavy metal ions in S/S matrix” is not sufficient and should be re-written with a proper explanation.

11. Authors should also write about the limitations of the approaches presented in the paper.

12. The conclusion section is particularly clear, plain, and simple. The conclusion is not concise. Need to be revised.

13. The authors need to take notes in the revision stage and cite relevant references including high-impact journals to make the manuscript to a broad range of readers.

14. Add some recent references and also, the numbers of references need to be extended in the revision stage.

Author Response

See the comments below:

  1. Paper title needs to revise.

Authors’ reply:

Thanks for your suggestion. The title has been revised as “Review on the interactions between conventional cementitious materials and heavy metal ions in the stabilization/solidification processing”.

  1. Abstract: Need to revise. In the abstract, add a description to clarify the flow of work.

Authors’ reply:

Thanks for your suggestion. The revision has been done as follow:

In the past few decades, the solidification/stabilization (S/S) technology has been put forward for the purpose of improving the soil strength and inhibiting the contaminant migration in the remediation of heavy metal contaminated sites. Cement, lime and fly ash are among the most commonly and effective binders to treat contaminated soils. During S/S processing, the main interactions that responsible for improving the soil behaviors can be concluded as gelification, self-hardening and aggregation. Currently, precipitation, incorporation and substitution have been commonly accepted as the predominant immobilization mechanisms for heavy metal ions, and have been directly verified by some micro-testing techniques. While replacement Ca2+/Si4+ in the cementitious products and physical encapsulation still remains controversial, which is proposed depending on the indirectly results. Lead and zinc can retard both the initial and final setting time of cement hydration, while chromium can accelerate the initial cement hydration. Though cadmium can shorten the initial setting time, the further cement hydration will be inhibited. While for the mercury, the interference impact is closely associated to its adapted anion. It should be pointed out that obtaining a better understanding of the remediation mechanism involved in S/S processing will facility the technical improvement, further extension and application.

  1. Authors are advised to revise the abstract and please focus abstract on the review approach.

Authors’ reply:

Thanks for your suggestion. The revision has been done as follow:

In the past few decades, the solidification/stabilization (S/S) technology has been put forward for the purpose of improving the soil strength and inhibiting the contaminant migration in the remediation of heavy metal contaminated sites. Cement, lime and fly ash are among the most commonly and effective binders to treat contaminated soils. During S/S processing, the main interactions that responsible for improving the soil behaviors can be concluded as gelification, self-hardening and aggregation. Currently, precipitation, incorporation and substitution have been commonly accepted as the predominant immobilization mechanisms for heavy metal ions, and have been directly verified by some micro-testing techniques. While replacement Ca2+/Si4+ in the cementitious products and physical encapsulation still remains controversial, which is proposed depending on the indirectly results. Lead and zinc can retard both the initial and final setting time of cement hydration, while chromium can accelerate the initial cement hydration. Though cadmium can shorten the initial setting time, the further cement hydration will be inhibited. While for the mercury, the interference impact is closely associated to its adapted anion. It should be pointed out that obtaining a better understanding of the remediation mechanism involved in S/S processing will facility the technical improvement, further extension and application.

  1. Some keywords should be revised in the manuscript.

Authors’ reply:

Thanks for your suggestion. The keywords has been revised as: “heavy metal contaminated soils; stabilization/solidification; remediation mechanisms; ions immobilization mechanisms; interference impacts”.

  1. The manuscript is in itself very hard to read. Several instances of repeated sentences have been used to describe the same thing. This has to be extensively taken care of throughout the manuscript.

Authors’ reply:

Thanks for your suggestion. The writing of this manuscript has been rechecked by a paid editing service offered by MDPI, and further revised by the authors.

The follow is the certification of English-editing.

  1. There are multiple grammatical errors in the manuscript. The manuscript could utilize some improvement in English by checking it with a native English speaker or software which provides their services for improving the quality of the manuscript.

Authors’ reply:

Thanks for your suggestion. The writing of this manuscript has been rechecked by a paid editing service offered by MDPI, and further revised by the authors.

The follow is the certification of English-editing.

  1. The manuscript also provides a very general overview of the topic. The review should include more critical inputs including comparisons between various literature and a critical assessment of the approach’s pros and cons.

Authors’ reply:

Thanks for your suggestion.

This paper reviewed the typically physicochemical reactions of cement, lime and fly ash, which are responsible for the S/S treatment. And the governed mechanisms for each binders were determined by referring to a great deal of literature, as summarized in Figure 1. Then in section 3 and 4, the authors presented the conclusions that were accepted by most researchers, which referred to the immobilization mechanisms of different heavy metal ions and the interferential impacts of contaminant on the performance of S/S process. Subsequently, the controversial results obtained by some individual researchers were also discussed. Finally, the comparison of these findings were listed in Table 3.

  1. The introduction section needs to be revised. Some more recent developments in this field are required to incorporate into the introduction section.

Authors’ reply:

Thanks for your suggestion. The revision has been done as follow:

“In the past few decades, the scale of the contaminated sites increased by years in China with the rapid development of industrialization and urbanization. In order to stop the further extension of soil pollution, a variety of environmental policies have been oper-ated since the initial of 12th Five-Year Plan. Moreover, the financial input was estimated to be increased to trillions in 2020 to encourage the technological innovation and improve-ment in the remediation of contaminated sites [1]. On this basis, quite a few technologies are proposed, and subsequently proved to be available to treat heavy metal contaminated soils, such as physical separation, incineration, solidification/stabilization (S/S), thermal desorption and so on [2-5]. Among the various innovative technologies, the S/S method has been extensively adopted and investigated. For instances, the Annual Report on Site Remediation Technologies (ASR, the 12th edition) published that S/S method was used in 217 of 977 sites, which were financially supported by the U.S. Superfund between 1982 and 2005. While in China, S/S has also been increasingly used in the remediation projects of heavy metal contaminated sites. This has been the case due to……”

  1. Figure-1 quality needs to improve.

Authors’ reply:

Thanks for your suggestion. The quality of Figure 1 has been improved as follows:

  1. The explanation of the section “3. Immobilization mechanisms of heavy metal ions in S/S matrix” is not sufficient and should be re-written with a proper explanation.

Authors’ reply:

Thanks for your suggestion. The relevant result obtained from the previous studies performed by the authors has been added in the manuscript to supplement the explanation and made the comparison with the work of the other researchers. For instance:

3.1 lead

“…Additionally, in the alkaline environment that is offered by cement hydration, Pb2+ is ready to precipitate as a lead hydroxide and lead carbonate, or occasionally as a lead sulfate, a lead hydroxyl carbonate (PbSO4, 3PbCO3•2Pb(OH)2•H2O), and a lead carbonate sulfate hydroxide (Pb4SO4(CO3)2(OH)2) [10, 71]. According to the logarithm of the stability constants of the formation of lead hydroxide and carbonate (10.9 and 13.1, respectively), lead carbonate appears to be more preferentially formed than that of hydroxide [72, 73] solids…”

“.All the above mentioned immobilization forms of lead were simultaneously detected by X-Ray Absorption Fine Structure spectroscopy and Raman spectroscopy, which was per-formed by Contessi et al. [16]. While incorporation into C-S-H structure is the most pro-nounced comparing to the other lead immobilization mechanisms…”

3.2 zinc

“…A calcium zincate hydrate (CaZn2(OH)6•2H2O), other than C-S-H and Ca(OH)2, precipitat-ed at the interface between the tri-calcium silicate paste (C3S) and the zinc wire was ob-served by Tashiro and Tatibano [87] and Lo et al. [88]. And, Liu et al. [7] also observed the presence of both zinc hydroxide and calcium zincate hydrate, which were responsible for the zinc immobilization at a high level of zinc concentration. Mollah et al. [89] made a further…”

“…Ziegler et al. [93] proposed that Zn2+ can be incorporated into the C-S-H structure by form-ing a Si-O-Zn bond. Liu et al. [94] reported that CSH containing Zn2+ was detected in a S/S treated soil by SEM test equipped with EDS. It is speculated that Zn2+ is immobilized by incorporation into the interlayer of CSH under a lower Zn2+ concentration condition. Rose et al. [95] illustrated this binding process based on EXAFS and 29NMR analysis, which…”

  1. Authors should also write about the limitations of the approaches presented in the paper.

Authors’ reply:

Thanks for your suggestion. The revision has been done as follow:

“Currently, the most commonly used technologies to treat heavy metal contaminated sites include solidification/stabilization, electro-kinetic technology, soil washing, ceramic solidification and phytoremediation, etc. Comparing to the other technologies, S/S is a mature construction technology characterized by the excellent effectiveness of remediation, low-cost and short duration. The cost and duration for each technologies are listed in Table 5.

Besides the above mentioned advantages, some disadvantages are still outstanding for S/S technology. The commonly used binders are almost silicate materials, the hardening paste of which has a poor durability when exposed to the sulfate attack. Otherwise, the acid rain is extensively distributed in China, which imposes the stabilized contaminant to face the risk of re-filtration.”

  1. The conclusion section is particularly clear, plain, and simple. The conclusion is not concise. Need to be revised.

Authors’ reply:

Thanks for your suggestion. The conclusions has been revised as follows:

“During the rapid development of industrialization and urbanization, S/S technology has also been increasingly used in the remediation projects of heavy metal contaminated sites. Dur-ing S/S processing, the main interactions that responsible for improving the soil behaviors can be concluded as precipitation, chemical incorporation, isomorphous substitution, physical adsorption and encapsulation. Currently, precipitation, incorporation and substitution have been commonly accepted as the predominant immobilization mechanisms for heavy metal ions, and have been directly verified by some micro-testing techniques. While re-placement Ca2+/Si4+ in the cementitious products and physical encapsulation still remains controversial, which is proposed depending on the indirectly results. Lead and zinc can retard both the initial and final setting time of cement hydration, while chromium can ac-celerate the initial cement hydration. Though cadmium can shorten the initial setting time, the further cement hydration will be inhibited. While for the mercury, the interference im-pact is closely associated to its adapted anion. Noted that the complex environment of S/S system (such as the contaminant concentration, pH level, molar ratio of Ca/Si and Ca/Al) makes it difficult to identify the controlling immobilization mechanism for a specific contaminant.

In general, the s/s treated heavy metal contaminated soil with conventional binders appears to be poor durability when exposed to the complicated and changeable environment, such as drying-wetting cycles, freezing-thawing cycles, acid rain leaching condition, chloride and sulfate attack. Otherwise, the manufacture of the conventional binder is always accompanied by the emission of greenhouse gases, which is actually opposite to the environmental protection policy in China. Thus, there is a really pressing need to research and develop the novel binders that can replace the conventional ones and have the better performance. By this, obtaining a better understanding of the remediation mechanism involved in S/S processing is of great importance to facility the technical improvement, further extension and application of this technology.”

  1. The authors need to take notes in the revision stage and cite relevant references including high-impact journals to make the manuscript to a broad range of readers.

Authors’ reply:

Thanks for your suggestion. The relevant references published in the high-impact journals, such as Journal of hazardous materials, Chemosphere, Construction and building materials, Cement and concrete researches, Soils and Foundations, Science of the Total Environment, have been cited in this manuscript.

  1. Add some recent references and also, the numbers of references need to be extended in the revision stage.

Authors’ reply:

Thanks for your suggestion. The recent references have been added in the manuscript, and the numbers of references has been extended as well, which can be followed in the section of References.

Round 2

Reviewer 3 Report

Recommendation: Accept

This paper will be useful to readers and researchers working in this area. I recommend publishing the manuscript in its present form.